# Survival Benefit of Renin-Angiotensin System Blockers in Critically Ill Cancer Patients: A Retrospective Study

**DOI:** 10.3390/cancers15123183

**Published:** 2023-06-14

**Authors:** Driss Laghlam, Anis Chaba, Matthias Tarneaud, Julien Charpentier, Jean-Paul Mira, Frédéric Pène, Clara Vigneron

**Affiliations:** 1Service de Médecine Intensive-Réanimation, Hôpital Cochin, Assistance Publique-Hôpitaux de Paris (AP-HP), Centre—Université Paris Cité, 75014 Paris, France; driss.laghlam@aphp.fr (D.L.); anis.chaba@hotmail.fr (A.C.); matthias.tarneaud@aphp.fr (M.T.); julien.charpentier@aphp.fr (J.C.); jean-paul.mira@aphp.fr (J.-P.M.); frederic.pene@aphp.fr (F.P.); 2Université Paris Cité, 75014 Paris, France; 3Inserm U1016, CNRS UMR8104, Institut Cochin, 75014 Paris, France

**Keywords:** renin-angiotensin-aldosterone system inhibitors, cancer, prognosis, critical care, angiotensin receptor blockers, angiotensin-converting enzyme inhibitors

## Abstract

**Simple Summary:**

The involvement of the renin-angiotensin pathway in both the regulation of the cardiovascular system and in tumorigenesis raises the question of the prognostic impact of renin-angiotensin system blockers (RABs) in cancer patients experiencing life-threatening complications. The aim of our retrospective study was to assess this impact in solid tumor patients requiring unplanned ICU admission over a 14-year period. Among 1845 patients mainly diagnosed with gastrointestinal and lung cancers, 414 (22.4%) were treated with RABs: 220 (53.1%) with angiotensin-receptor blockers (ARBs) and 194 (46.9%) with angiotensin-converting enzyme inhibitors (ACEis). ARBs use and ACEis use were both associated with improved in-ICU survival, whereas only ARBs use was associated with improved one-year survival.

**Abstract:**

Increasing evidence argues for the promotion of tumorigenesis through activation of the renin-angiotensin system pathway. Accordingly, a benefit of renin-angiotensin system blockers (RABs) treatments has been suggested in patients with solid cancers in terms of survival. We aimed to evaluate in-ICU survival and one-year survival in cancer patients admitted to the ICU with respect to the use of RABs. We conducted a retrospective observational single-center study in a 24-bed medical ICU. We included all solid cancer patients (age ≥ 18 years) requiring unplanned ICU admission. From 2007 to 2020, 1845 patients with solid malignancies were admitted (median age 67 years (59–75), males 61.7%). The most frequent primary tumor sites were the gastrointestinal tract (26.8%), the lung (24.7%), the urological tract (20.1%), and gynecologic and breast cancers (13.9%). RABs were used in 414 patients, distributed into 220 (53.1%) with angiotensin-receptor blockers (ARBs) and 194 (46.9%) with angiotensin-converting enzyme inhibitors (ACEis). After multivariate adjustment, ARBs use (OR = 0.62, 95%CI (0.40–0.92), *p* = 0.03) and ACEis use (OR = 0.52, 95%CI (0.32–0.82), *p* = 0.006) were both associated with improved in-ICU survival. Treatment with ARBs was independently associated with decreased one-year mortality (OR = 0.6, 95%CI (0.4–0.9), *p* = 0.02), whereas treatment with ACEis was not. In conclusion, this study argues for a beneficial impact of RABs use on the prognosis of critically ill cancer patients.

## 1. Introduction

Renin-angiotensin blockers (RABs), namely angiotensin-receptor blockers (ARBs) and angiotensin-converting enzyme inhibitors (ACEis), are well known for their cardiovascular protective properties. The impact of pre-admission RABs on the outcome of critically ill patients remains controversial. RABs may theoretically worsen hemodynamic and renal failures, but the requirements for vasopressors of RAB-treated septic patients were not modified compared to septic patients without anti-hypertensive medication or patients treated with beta-blockers or calcium channel blockers [1]. Hence, prior use of RABs did not result in increased mortality [2] and, conversely, likely improved survival in general intensive care unit (ICU) populations [3,4]. In ICU survivors who experienced acute kidney injury, RABs prescription after ICU discharge was associated with a decrease in one-year mortality [5], possibly linked to reduced incidence of major cardiac adverse events [6]. In a recent study by Angriman et al. reporting the outcomes of 7174 sepsis survivors, a new treatment with RABs within 30 days of hospital discharge was associated with a reduction in major cardiovascular events (i.e., myocardial infarction, stroke) compared to new users of a calcium channel blocker or a thiazide diuretic [7]. Besides, there is increasing evidence that the renin-angiotensin system pathway may drive malignant tumor progression through inhibition of apoptosis associated with enhanced tumor proliferation, cell migration, tissue invasion, and angiogenesis [8].

Patients with cancer account for a growing proportion of ICU admissions as one in eight patients admitted to the ICU has a solid tumor, as reported in a multicenter observational study including 3147 patients from 24 European countries [9]. An international study performed in 84 countries described that 12.1% of critically ill patients had a history of solid cancer [10]. The lifetime probability of being diagnosed with invasive cancer is slightly higher for men (40.2%) than for women (38.5%), reflecting life expectancy as well as cancer risk [11]. It is the leading cause of death among women aged 40 to 79 years and men aged 60 to 79 years in the United States [11]. Besides the stage of malignancy, comorbidities play an important role in the prognosis of cancer patients. Hypertension, sometimes linked to side effects of antitumoral treatment (particularly anti-VEGF agents) [12], is the more frequent non-malignant comorbid condition in patients with cancer, affecting 38% of these individuals [13]. Recent hypertension practice guidelines published in 2020 recommend RABs as a first-line treatment [14]. Hence, a survival benefit of renin-angiotensin system blockers treatments has been suggested in patients with solid cancers [15,16,17], through reduction in cancer treatment–related adverse events, such as chemotherapy-induced cardiotoxicity, arterial hypertension and radiation injury [18,19], or possibly through inhibition of tumorigenesis [8]. 

The involvement of the renin-angiotensin pathway in both the regulation of the cardiovascular system and in oncogenesis raises the question of the prognostic impact of RABs in cancer patients experiencing life-threatening complications. The objective of the present study was to address the impact of RABs treatments on short-term and long-term outcomes in patients with solid malignancies requiring ICU admission.

## 2. Materials and Methods

In this single-center retrospective study in a 24-bed medical ICU, we enrolled all adult patients (age ≥18 years) with a diagnosis of solid neoplasm (before or during the ICU stay) requiring unplanned admission from January 2007 to December 2020. Exclusion criteria were cancer cured for more than 5 years, planned admissions after elective surgery, and admissions for securing a procedure. Data were extracted from the patients’ data management system (Clinisoft^®^, GE Healthcare, Chicago, IL, USA) and computed from the individual medical file. We collected the following data: demographic characteristics (age, sex), comorbidities (hypertension, diabetes mellitus, cirrhosis, chronic renal failure, chronic dialysis, chronic respiratory failure, ischemic cardiopathy, chronic heart failure, peripheral arterial disease, and stroke), and prior chronic medications including ACEis and ARBs. Concerning the underlying malignancy, the date of diagnosis, primary tumor site (lung, breast, gastrointestinal, urologic, skin, gynecologic, head and neck, others), cancer stage (localized, advanced, metastatic), status of response to treatment (newly diagnosed, partial remission, complete remission, progression), and recent oncological treatment within the last 3 months (surgery or chemotherapy) were recorded. Regarding the ICU stay, we recorded admission Sequential Organ Failure Assessment (SOFA) score [20] and Simplified Acute Physiology Score (SAPS) II [21], organ failure supports (invasive mechanical ventilation, non-invasive mechanical ventilation, vasopressors/inotropes, and renal replacement therapy), and decisions to forgo life-sustaining therapies (DFLSTs). Missing data were handled using the Random Forest method. Admissions directly linked to the underlying malignancy were classified as cancer-specific complications [22]. The main outcomes were in-ICU and in-hospital survival in the whole cohort and one-year survival in ICU survivors with complete follow-up.

Variables were reported as absolute value with count and percentage for categorical variables or median with interquartile range for quantitative variables. Univariate analyses were performed using the χ^2^ or Fisher exact test, and the non-parametric Mann–Whitney and Kruskal–Wallis tests, as appropriate. A *p*-value < 0.05 was considered significant. Multivariate models used a conditional backward stepwise variable selection process based upon variable influence in univariate analysis. Critical entry and exit p values were 0.2 and 0.1, respectively. Logistic regression was performed to identify the possible factors involved in ICU and one-year mortality. Kaplan–Meier survival curves were performed to illustrate data found in the logistic regression. Correlation and interaction were checked within final models. Therefore, DSFLTs were excluded from the ICU mortality model due to a significant interaction with SAPS2 score. We also explored the impact of RABs on one-year mortality, with a propensity score weighting in ICU survivors. Variables included for the propensity score models were age, cancer status and stage, gender, hypertension, diabetes, chronic renal insufficiency, lung cancer. Statistical significance was considered using two-sided tests with a critical alpha risk of 0.05. Statistical analyses were performed using R version 4.2.1 (R Foundation for Statistical Computing), “survival”, “survey”, “WeightIt”, and “PSweight” packages. 

According to French regulations, this study was approved by the ethics committee of the French Intensive Care Society (Société de Réanimation de Langue Française, CE SRLF #17-03), which waived the need for signed consent.

## 3. Results

### 3.1. Baseline Characteristics 

From 2007 to 2020, 1845 patients with underlying solid neoplasms were admitted to the ICU. Baseline characteristics of the whole cohort are displayed in Table 1. The median age was 67 years (59–75) and 61.7% were men. The most frequent primary tumor sites were the gastrointestinal tract (*n* = 494, 26.8%), lung (*n* = 456, 24.7%), the urological tract (*n* = 371, 20.1%), and gynecologic and breast cancers (*n* = 256, 13.9%). Malignancy was at metastatic stage in 476 patients (49.2%). Features at ICU admission and outcomes are displayed in Table 2. The median SOFA score at ICU admission was five points (4–7). The main cause of admission was infection (*n* = 650, 35.2%), followed by specific complications directly linked to the underlying malignancy (*n* = 488, 26.4%). Of the patients, 959 (51.9%) required invasive mechanical ventilation, 632 (34.2%) vasopressor/inotrope support, and 303 (16.4%) renal replacement therapy. Life-supporting therapies were withheld or withdrawn in 529 (28.6%) patients. The in-ICU, in-hospital, and one-year mortality rates were 22.1%, 41.7%, and 66.6%, respectively.

RABs were used in 414 patients, including 220 (53.1%) with ARBs and 194 (46.9%) with ACEis. RABs users were older (72 (65–78) vs. 66 (57–74) years, *p* < 0.001) and more frequently had diabetes (155 (37.4%) vs. 197 (13.8%), *p* < 0.001), chronic renal failure (65 (15.7%) vs. 100 (7.0%), *p* < 0.001) and chronic heart failure defined as left ventricular ejection fraction <50% (45 (10.9%) vs. 43 (3.0%), *p* = 0.04). With regard to cancer status, RABs users less frequently had metastatic malignancies or progression under treatment (respectively, 195 (47.1%) vs. 809 (56.5%), *p* = 0.001 and 114 (27.5%) vs. 532 (37.2%) *p* = 0.006). RABs users were more often admitted for acute renal failure (41 (9.9%) vs. 97 (6.8%), *p* = 0.043) and less often for a specific complication (80 (19.3%) vs. 408 (28.5%), *p* < 0.001).

### 3.2. ICU Mortality

Thirty-day survival curves according to the use of ARBs or ACEis are displayed in Figure 1 (log-rank, *p* = 0.074). After adjustment in a multivariate model, ARBs use (OR = 0.62, 95%CI (0.40–0.92), *p* = 0.03) and ACEis use (OR = 0.52, 95%CI (0.32–0.82), *p* = 0.006) were both associated with improved in-ICU survival (Figure 2, Panel A). Other independent determinants of in-ICU mortality were as follows: SAPS2 score, lung cancer, metastatic stage, newly diagnosed status, and cancer progression under treatment.

### 3.3. In-Hospital Mortality

In-hospital mortality rates in no-RABs patients, ARBs users, and ACEis users were 42.8%, 35.5% and 38.3%, respectively (*p* = 0.09). In multivariate analysis, ARBs use was independently associated with decreased in-hospital mortality (OR = 0.65, 95%CI (0.44–0.97), *p* = 0.04) while ACEis use nearly reached significance (OR = 0.66, 95%CI (0.44–1.00), *p* = 0.06) (Figure 2, Panel B).

### 3.4. One-Year Mortality

Based on a logistic regression analysis among ICU survivors, chronic arterial hypertension (OR = 1.9, 95%CI (1.4–2.7), *p* < 0.001) and ARBs use (OR = 0.56, 95%CI (0.35–0.90), *p* = 0.02) were independently associated with one-year mortality but not ACEis use (OR = 0.81, 95%CI (0.5–1.3), *p* = 0.4) (Figure 2, Panel C). Other independent determinants of one-year mortality were as follows: decision to forgo life-sustaining therapy during ICU stay, lung cancer, SAPS2, newly diagnosed status, partial remission status, cancer in progression under treatment, advanced stage, metastatic stage, and specific cancer-related complications.

In order to fully explore the effects of ARBs use on one-year mortality, we performed a propensity score matching (using overlap weighting) based on ARBs use in the whole cohort (excluding ACEis users). After weighting, both groups and propensity scores were adequately balanced (Appendix A). One-year survival curves in the propensity-score matched cohorts of patients, treated or not with ARBs, are displayed in Figure 3. Compared to the patients without RABs, the patients with ARBs exhibited decreased one-year mortality (117/220 (60.0%) vs. 863/1431 (67.5%), *p* = 0.046).

## 4. Discussion

In this large cohort of cancer patients, we found that RABs users had an improved survival in the ICU. Although this association was observed after multivariate analysis, patients treated with RABs more frequently experienced rapidly reversible organ failures (i.e., renal failure) and were less hospitalized because of specific complications directly linked to the underlying malignancy.

Concerning long-term outcome, ARBs users had improved one-year survival. Of note, although hypertension was associated with a worse prognosis, ARBs users had a better one-year survival, suggesting a specific effect of this medication. Our findings align with recent clinical studies reporting improved prognosis in cancer patients treated with RABs as adjunctive therapy [15,23,24]. Interestingly, this benefit of using ARBs appears even in cancer patients who underwent an unplanned admission in ICU, constituting a specific population not previously described. Furthermore, improved survival was noted in a cohort of patients with various cancers, mainly gastrointestinal, urologic, lung, and breast cancers as previously describe [15,25,26,27].

The effects of RABs in cancer patients remain elusive because observational studies yield conflicting results. The meta-analyses and retrospective studies may have inherent bias as they were never designed to explore any pro- or antitumoral effects and included heterogeneous patients with various cancers at different stages. However, the accumulation of evidence of an improved survival linked to their plausible impact on tumorigenesis pathways, and the absence of significant side effects of these widely prescribed low-cost drugs, make the use of RABs possibly interesting in this population. Most importantly, the renin-angiotensin system consists of two main axes that function in opposition to each other. There is increasing evidence that activation of the classical renin-angiotensin system pathway through Angiotensin II-angiotensin-1-receptor (AT1R) drives tumor progression through enhanced tumor proliferation, migration, invasion and angiogenesis, as well as inhibition of apoptosis. In contrast, the Angiotensin (1,7)-Mas receptor and the Angiotensin II-AT2R pathways are thought to antagonize many of the cellular actions of the Angiotensin II-AT1R axis [8,28]. In detail, angiotensin converting enzyme 2 (ACE2) and neprilysin (NEP) are part of this pathway with opposite physiological effects. ACE2 and NEP can cleave Angiotensin II to produce angiotensin 1–7 which binds to Mas receptor, while ACE2 cleaves Angiotensin I to generate angiotensin 1–9 which activates AT2R pathway, mediating antiproliferative effects [29]. The specific anti-tumorigenesis effects of ACEis and ARBs are not well known. Both may exert their effect via inhibition of angiotensin II production and therefore by blocking the angiotensin II-AT1R pathway. Furthermore, with ACEis, ACE2 production of Angiotensin (1–9) is forced, activating the AT2R pathway. With ARBs, the circulating angiotensin II can be turned to Angiotensin (1–7) by the available ACE2, mediating antiproliferative effect. However, there are still many questions about the cellular mechanisms involved in the effects of these two molecules. In details, Angiotensin II was found to enhance tumor growth by increasing production of tumor-promoting macrophages [30]. Conversely, blockade of the renin-angiotensin system by angiotensin-converting enzyme inhibitors was associated with an increased expression of genes linked with the activity of T cells and antigen-presenting cells, and a longer survival independently of chemotherapy [31]. In ovarian cancer models, RABs (losartan) use normalizes tumor microenvironment via its antifibrotic effects, and through this improves vessel perfusion and drug delivery, enhancing chemotherapy efficacy [32]. Regarding clinical investigations, in a recent retrospective study, patients with hypertension who were concomitantly taking RABs during immune checkpoint inhibitors therapy had better overall survival, especially for those with gastrointestinal and genitourinary cancers, without any significant change regarding immune related adverse events [15]. The effects of RABs on survival could be driven by mitigating cancer-treatment-related adverse events, on top of inhibition of tumor growth and recurrence, as described in breast cancers in which RABs decrease the cardiotoxicity in patients treated with trastuzumab or anthracycline-containing regiment [19,33,34]. In our study, it is noteworthy that RABs users, compared to non-RABs users, displayed different baseline cancer characteristics with less advanced malignancies, which could affect long-term prognosis. Nevertheless, we used a propensity-score matching method to compare these patients.

We did not retrieve any association between use of ACEis and one-year mortality in this specific population. These observations were consistent with other studies as the effects of ACEis in cancer risk and survival are still debated [25,35]. In 2018, Hicks et al. found that long-term use of ACEis was specifically associated to an increased risk of lung cancer compared to ARBs in a population base cohort study of 992,061 patients newly treated with antihypertensive drugs [36]. This association could be linked to an accumulation of bradykinin, stimulating tumor cells proliferation and increasing vascular permeability, therefore facilitating a greater supply of nutrients and oxygen to the tumor cells [37]. Moreover, substance P induces the proliferation and migration of tumor cells and stimulates angiogenesis [38]. Further large, randomized studies are needed to fully explore the effects of RABs as adjunctive treatment in cancer patients.

We acknowledge several limitations. Although our results are consistent with current literature, this was a single-center retrospective study, and these results may not be fully transposable elsewhere, as our cohort reflects the case mix of cancer patients followed up in our institution. In addition, due to its retrospective design, some characteristics were missing, and others could not be reliably collected, including the dose and duration of RABs treatment especially after hospital discharge, the relevant oncologic outcomes, and the primary causes of death. Furthermore, this study analyzed patients hospitalized over a 14-year period, during which patients’ characteristics, ICU management, and prognosis have probably evolved. Finally, despite consistent results in the current literature, association is not causation so we cannot fully conclude on the benefit of these treatments in patients with malignancies.

## 5. Conclusions

In this retrospective cohort of patients with various solid malignancies requiring unplanned ICU admissions, ARBs users had a better one-year survival than non-RABs users, despite experiencing a life-threatening complication. We make the assumption that it might be linked to antiproliferative effect or decreased cancer-treatment adverse events, although it has to be explored in prospective studies. We did not find an association between ACEis use and one-year mortality in this population.

## Figures and Tables

**Figure 1 cancers-15-03183-f001:**
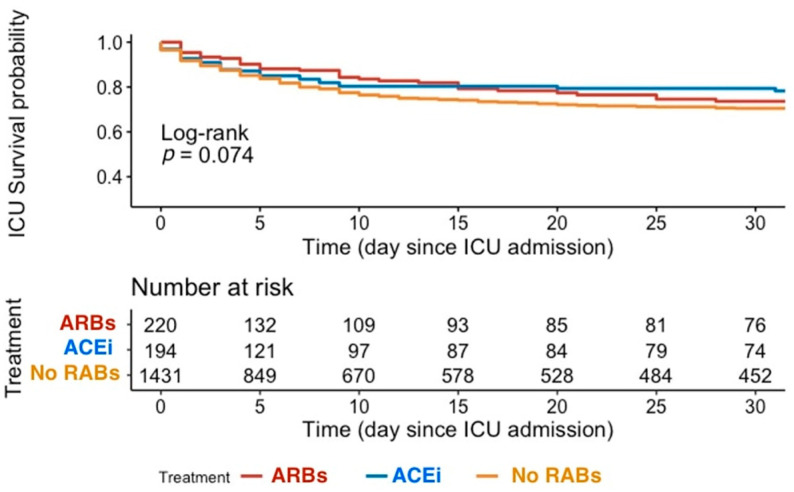
Kaplan–Meier survival estimates of ICU survival according to RABs use. Abbreviations: ACEis: angiotensin-converting enzyme inhibitors; ARBs: angiotensin-receptor blockers; ICU: intensive care unit; RABs: renin-angiotensin system blockers.

**Figure 2 cancers-15-03183-f002:**
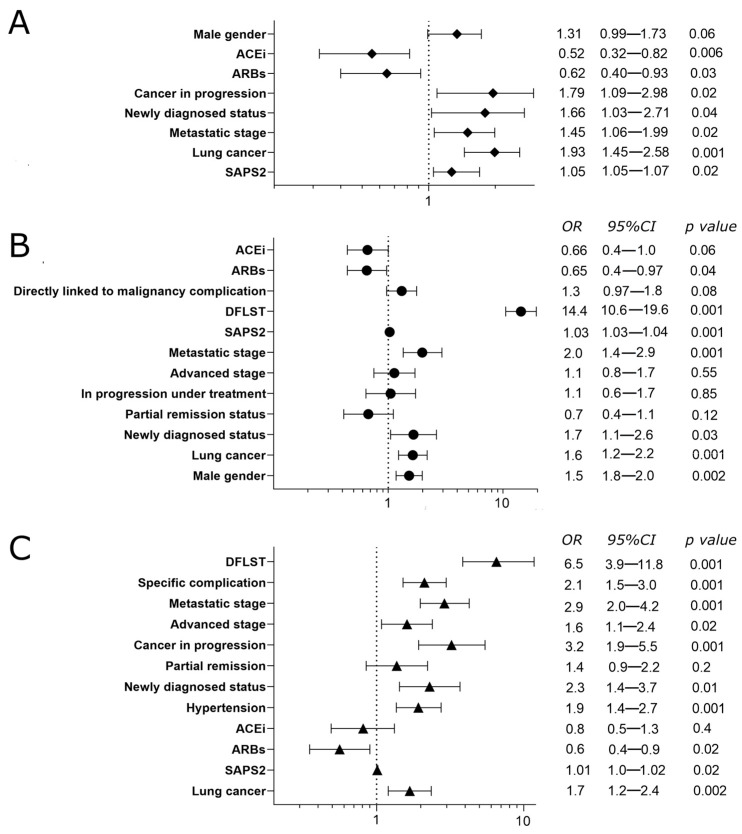
Independent determinants of in-ICU, in-hospital, and one-year mortality. Panel (**A**) Independent risk factors for in-ICU mortality. Panel (**B**) Independent risk factors for in-hospital mortality. Panel (**C**) Independent risk factors for one-year mortality. Abbreviations: ACEis: angiotensin-converting enzyme inhibitors; ARBs: angiotensin-receptor blockers; CI: confidence interval; DFLSTs: decisions to forgo life-sustaining therapies; ICU: intensive care unit; OR: Odds Ratio; SAPS II: Simplified Acute Physiology Score II.

**Figure 3 cancers-15-03183-f003:**
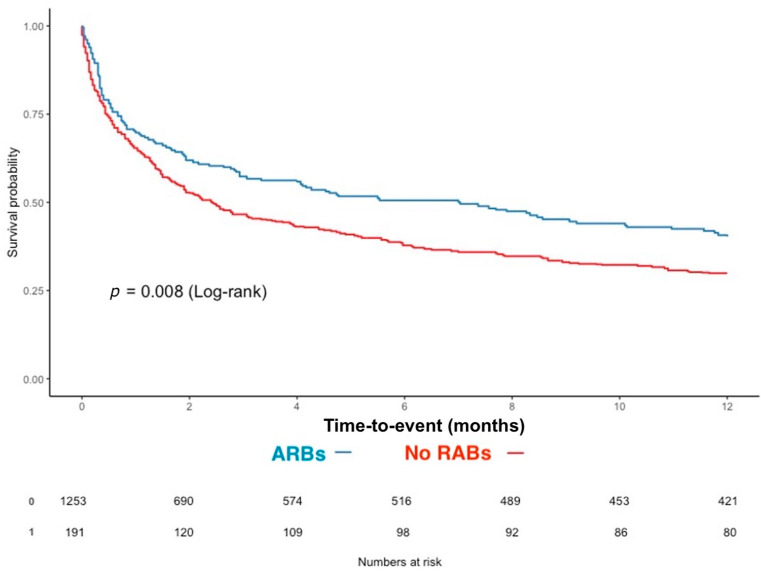
Kaplan–Meier survival estimates of one-year survival according to ARBs use after propensity score matching. Abbreviations: ARBs: angiotensin-receptor blockers; RABs: renin-angiotensin system blockers.

**Table 1 cancers-15-03183-t001:** Baseline patients’ characteristics.

	No RABs *n* = 1431	ARBs *n* = 220	ACEis *n* = 194	*p*
Demographic conditions				
Age, median [IQR]	66.0 [57.0–74.0]	72.0 [65.0–79.0]	72.0 [65.0–78.0]	<0.001
Male gender, *n* (%)	851 (59.5)	143 (65.0)	144 (74.2)	<0.001
Non-cancer comorbid conditions				
Hypertension, *n* (%)	382 (26.7)	220 (100.0)	194 (100.0)	<0.001
Diabetes mellitus, *n* (%)	197 (13.8)	85 (38.6)	70 (36.1)	<0.001
Cirrhosis, *n* (%)	85 (5.9)	12 (5.5)	15 (7.7)	0.569
Chronic renal failure *n*, (%)	100 (7.0)	38 (17.3)	27 (13.9)	<0.001
Chronic dialysis, *n* (%)	10 (0.7)	1 (0.5)	1 (0.5)	0.888
Chronic respiratory failure, *n* (%)	57 (4.0)	7 (3.2)	14 (7.2)	0.079
Ischemic cardiopathy, *n* (%)	121 (8.5)	47 (21.4)	58 (29.9)	<0.001
Chronic heart failure, *n* (%)	43 (3.0)	17 (7.7)	28 (14.4)	<0.001
Peripheral arterial disease, *n* (%)	56 (3.9)	19 (8.6)	30 (15.5)	<0.001
Stroke, *n* (%)	46 (3.2)	9 (4.1)	24 (12.4)	<0.001
Type of cancer, *n* (%)				0.056
Lung	353 (24.7)	56 (25.5)	47 (24.2)	
Breast	149 (10.4)	17 (7.7)	11 (5.7)	
Gastrointestinal	380 (26.6)	60 (27.2)	54 (27.8)	
Urologic	259 (18.1)	60 (27.3)	52 (26.8)	
Skin	27 (1.9)	3 (1.4)	3 (1.5)	
Gynecologic	63 (4.4)	8 (3.6)	8 (4.1)	
Head and neck	53 (3.7)	5 (2.3)	6 (3.1)	
Others	147 (10.3)	11 (5.0)	13 (6.7)	
Stage, *n* (%)				0.002
Localized	304 (21.2)	74 (33.6)	54 (27.8)	
Advanced	308 (21.5)	42 (19.1)	44 (22.7)	
Metastatic	809 (56.5)	101 (45.9)	94 (48.5)	
Unknown	10 (0.6)	3 (1.4)	2 (1.0)	
Current status, *n* (%)				0.015
Newly diagnosed	409 (28.6)	66 (30.0)	65 (33.5)	
Partial remission	290 (20.3)	43 (19.5)	50 (25.8)	
Complete remission	182 (12.7)	40 (18.2)	30 (15.5)	
Progression	532 (37.2)	67 (30.5)	47 (24.2)	
Unknown	18 (1.3)	4 (1.8)	2 (1.0)	
Recent oncological treatment (<3 months)				
Chemotherapy <3 months	734 (51.4)	96 (43.6)	96 (49.5)	0.098
Surgery <3 months	205 (14.3)	40 (18.2)	27 (13.9)	0.309

Abbreviations: ACEis: angiotensin-converting enzyme inhibitors; ARBs: angiotensin-receptor blockers; RABs: renin-angiotensin system blockers.

**Table 2 cancers-15-03183-t002:** Features at ICU admission and outcomes.

	No RABs *n* = 1431	ARBs *n* = 220	ACEis *n* = 194	
SOFA score, median [IQR]	5.0 [4.0–8.0]	5.0 [4.0–8.0]	5.0 [4.0–8.0]	0.861
SAPS2, median [IQR]	48.0 [35.0–63.0]	49.5 [40.0–66.8]	50.0 [38.0–68.0]	0.015
Reasons for admission				0.085
Cancer-specific complication	408 (28.5)	37 (16.8)	43 (22.2)	<0.001
Bleeding	69 (4.8)	19 (8.6)	10 (5.1)	
Infection	510 (35.6)	80 (36.4)	60 (30.9)	
Acute renal failure	94 (6.6)	24 (10.9)	17 (8.8)	
Ischemic event	16 (1.1)	1 (0.5)	6 (3.1)	
Thrombotic event	42 (2.9)	8 (3.6)	5 (2.6)	
Metabolic	84 (5.9)	9 (4.1)	7 (3.6)	
Others	502 (35.1)	65 (29.5)	70 (36.1)	
Organ failure supports				
Mechanical ventilation	735 (51.4)	110 (50.0)	114 (58.8)	0.126
Vasopressor/inotropes	476 (33.3)	80 (36.4)	76 (39.2)	0.208
Renal replacement therapy	215 (15.0)	51 (23.2)	37 (19.1)	0.006
Outcomes				
ICU mortality	332 (23.2)	42 (19.1)	34 (17.5)	0.104
In-hospital mortality	590/1370 (43.0)	76/214 (35.5)	72/187 (38.5)	0.087
6-month mortality	755/1295(58.3)	96/197 (48.7)	101/175 (57.7)	0.040
One-year mortality	863/1278 (67.5)	117/195 (60.0)	115/172 (66.9)	0.116
Decision to forgo life-sustaining therapy	419 (29.3)	56 (25.5)	54 (27.8)	0.487

Abbreviations: ACEis: angiotensin-converting enzyme inhibitors; ARBs: angiotensin-receptor blockers; ICU: intensive care unit; RABs: renin-angiotensin system blockers; SAPS II: Simplified Acute Physiology Score II; SOFA: Sequential Organ Failure Assessment.

## Data Availability

The datasets used and/or analyzed during the current study are available from the corresponding author on reasonable request.

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
