# Peer review of "Survival Benefit of Renin-Angiotensin System Blockers in Critically Ill Cancer Patients: A Retrospective Study"

_cancers, 2023, doi:10.3390/cancers15123183_

Round 1

Reviewer 1 Report

Remarks to the author:

In this particular study, the authors examined the one-year survival in cancer patients admitted to the ICU, focusing on the usage of angiotensin receptor blockers (ARBs). The investigation was conducted as a retrospective observational single-center study in a 24-bed medical ICU. The findings indicated that out of the 414 patients analyzed, 220 (53.1%) were treated with angiotensin receptor blockers (ARBs) and 194 (46.9%) received angiotensin-converting enzyme inhibitors (ACEi). According to their results, the use of ARBs was found to be independently associated with a reduction in one-year mortality (OR=0.6, 95% CI: 0.4-0.9, p=0.02). Conversely, no such association was observed with the use of ACEi. Consequently, the study concludes that among patients with solid malignancies admitted to the ICU, the administration of ARBs is linked to a more favorable prognosis within the ICU, and specifically, the use of ARBs is associated with improved one-year survival compared to ACEi.

Specific comments:

i)  The authors have asserted that the renin-angiotensin system (RAS) plays a role in driving malignant tumor progression by inhibiting apoptosis, thereby promoting tumor proliferation, cell migration, tissue invasion, and angiogenesis. However, it is important to note that the RAS consists of two main axes that function in opposition to each other. Therefore, the authors should elaborate on the specific axis of the RAS that contributes to these malignancies.

ii) Figure 3 - Authors need to explain the reason for comparing patients without RABs, patients with ARBs instead of comparing the same treatment received groups.

iii)   The discussion lacks a clear explanation for the differential effectiveness between angiotensin receptor blockers (ARBs) and angiotensin-converting enzyme inhibitors (ACEi) in reducing cancer progression.

Minor English editing is required.

Author Response

In this particular study, the authors examined the one-year survival in cancer patients admitted to the ICU, focusing on the usage of angiotensin receptor blockers (ARBs). The investigation was conducted as a retrospective observational single-center study in a 24-bed medical ICU. The findings indicated that out of the 414 patients analyzed, 220 (53.1%) were treated with angiotensin receptor blockers (ARBs) and 194 (46.9%) received angiotensin-converting enzyme inhibitors (ACEi). According to their results, the use of ARBs was found to be independently associated with a reduction in one-year mortality (OR=0.6, 95% CI: 0.4-0.9, p=0.02). Conversely, no such association was observed with the use of ACEi. Consequently, the study concludes that among patients with solid malignancies admitted to the ICU, the administration of ARBs is linked to a more favorable prognosis within the ICU, and specifically, the use of ARBs is associated with improved one-year survival compared to ACEi.

Specific comments:

  1. i) The authors have asserted that the renin-angiotensin system (RAS) plays a role in driving malignant tumor progression by inhibiting apoptosis, thereby promoting tumor proliferation, cell migration, tissue invasion, and angiogenesis. However, it is important to note that the RAS consists of two main axes that function in opposition to each other. Therefore, the authors should elaborate on the specific axis of the RAS that contributes to these malignancies.:

We added explanations in the discussion section on the two opposite renin-angiotensin axes and have clarified that it is the classic renin-angiotensin system through the activation of AngII-ATR1 which is mainly responsible for the pro-tumorogenic effect: “Most importantly, the renin angiotensin system consists of two main axes that function in opposition to each other. There is increasing evidence that activation of the classical renin-angiotensin system pathway through Angiotensin II-angiotensin-1-receptor (AT1R) drives tumor progression through enhanced tumor proliferation, migration, invasion and angiogenesis as well as inhibition of apoptosis. In contrast, the Angiotensin (1,7)-Mas receptor and the Angiotensin II-AT2R pathways are thought to antagonize many of the cellular actions of the Angiotensin II-AT1R axis [8,28]. In details, angiotensin converting enzyme 2 (ACE2) and neprilysin (NEP) are part of this pathway with opposite physiological effects. ACE2 and NEP can cleave Angiotensin II to produce angiotensin 1–7 which binds to Mas receptor, while ACE2 cleaves Angiotensin I to generate angiotensin 1–9 which activates AT2R pathway, mediating antiproliferative effects [29].”

  1. ii) Figure 3 - Authors need to explain the reason for comparing patients without RABs, patients with ARBs instead of comparing the same treatment received groups.

Thank you for your comment. Given that the one-year mortality of patients receiving ACEi and those not receiving RABs is comparable, and to confirm the effects of ARBs, we have chosen to compare two matched cohorts of patients: those receiving ARBs and those not receiving RABs.

iii)   The discussion lacks a clear explanation for the differential effectiveness between angiotensin receptor blockers (ARBs) and angiotensin-converting enzyme inhibitors (ACEi) in reducing cancer progression.

We agree with the reviewer and added further explications:

“The specific anti-tumorigenesis effects of ACEi and ARBs are not well known. Both may exert their effect via inhibition of angiotensin II production and therefore by blocking the angiotensin II-AT1R pathway. Furthermore, with ACEi, ACE2 production of Angiotensin(1-9) is forced, activating the AT2R pathway. With ARBs, the circulating angiotensin II can be turned to Angiotensin(1-7) by the available ACE2, mediating antiproliferative effect. However, there are still many questions about the cellular mechanisms involved in the effects of these two molecules.”

Comments on the Quality of English Language

Minor English editing is required

We tried to improve the quality of our manuscript

Reviewer 2 Report

This is an interesting although retrospective one center study whether renin-angiotensin system blockers can help survival of patients with solid cancers.  ICU survival and one-year survival of cancer patients admitted to the ICU were analyzed with respect to the use of RABs. It was a retrospective study, with 1845 patients.

The most frequent primary tumor sites were the gastrointestinal tract (26.8%), the lung (24.7%), the urological tract (20.1%), and gynecologic and breast cancers (13.9%). RABs were used in 414 patients, 220  with angiotensin-receptor blockers (ARBs) and 194 with angiotensin-converting enzyme inhibitors (ACEi). 

ARBs and ACEi were both associated with improved in-ICU survival. ARBs showed decreased one-year mortality.

The whole study seems well designed and aimed, the manuscript is well written.

Few small objections:

-The title seems not enough informative and attractive, possibly better if not a question

-Last lines in the abstract repeat what is already said few lines before. Some limitations of a retrospective study might use that space.

-In the discussion section alternative receptors of Ang II are mentioned. A casual reader might be interested in the ACE2 branch of the RAS system. With ACEi we force the ACE2 production of Ang(1-9) and limit the availability of dilatory Ang(1-7). With ARBs the circulating Ang II can be turned to Ang(1.7) by the available ACE2. The reader might be interested whether this difference might be relevant to cancer growth.

Author Response

This is an interesting although retrospective one center study whether renin-angiotensin system blockers can help survival of patients with solid cancers.  ICU survival and one-year survival of cancer patients admitted to the ICU were analyzed with respect to the use of RABs. It was a retrospective study, with 1845 patients. The most frequent primary tumor sites were the gastrointestinal tract (26.8%), the lung (24.7%), the urological tract (20.1%), and gynecologic and breast cancers (13.9%). RABs were used in 414 patients, 220  with angiotensin-receptor blockers (ARBs) and 194 with angiotensin-converting enzyme inhibitors (ACEi). ARBs and ACEi were both associated with improved in-ICU survival. ARBs showed decreased one-year mortality.The whole study seems well designed and aimed, the manuscript is well written.

Few small objections:

-The title seems not enough informative and attractive, possibly better if not a question

We modified the title as required: “Survival benefit of renin-angiotensin system blockers in critically ill cancer patients: a retrospective study”

-Last lines in the abstract repeat what is already said few lines before. Some limitations of a retrospective study might use that space.

We deleted this line and added a new one: “In conclusion, this study argues for a beneficial impact of RABs use on the prognosis of critically ill cancer patients.” 

-In the discussion section alternative receptors of Ang II are mentioned. A casual reader might be interested in the ACE2 branch of the RAS system. With ACEi we force the ACE2 production of Ang(1-9) and limit the availability of dilatory Ang(1-7). With ARBs the circulating Ang II can be turned to Ang(1.7) by the available ACE2. The reader might be interested whether this difference might be relevant to cancer growth

We agree with the reviewer and added further explications:

Most importantly, the renin angiotensin system consists of two main axes that function in opposition to each other. There is increasing evidence that activation of the classical renin-angiotensin system pathway through Angiotensin II-angiotensin-1-receptor (AT1R) drives tumor progression through enhanced tumor proliferation, migration, invasion and angiogenesis as well as inhibition of apoptosis. In contrast, the Angiotensin (1,7)-Mas receptor and the Angiotensin II-AT2R pathways are thought to antagonize many of the cellular actions of the Angiotensin II-AT1R axis [8,28]. In details, angiotensin converting enzyme 2 (ACE2) and neprilysin (NEP) are part of this pathway with opposite physiological effects. ACE2 and NEP can cleave Angiotensin II to produce angiotensin 1–7 which binds to Mas receptor, while ACE2 cleaves Angiotensin I to generate angiotensin 1–9 which activates AT2R pathway, mediating antiproliferative effects [29]. The specific anti-tumorigenesis effects of ACEi and ARBs are not well known. Both may exert their effect via inhibition of angiotensin II production and therefore by blocking the angiotensin II-AT1R pathway. Furthermore, with ACEi, ACE2 production of Angiotensin(1-9) is forced, activating the AT2R pathway. With ARBs, the circulating angiotensin II can be turned to Angiotensin(1-7) by the available ACE2, mediating antiproliferative effect. However, there are still many questions about the cellular mechanisms involved in the effects of these two molecules.”

Reviewer 3 Report

The authors have to highlight the limitations of their study and must include possible causes for the decrease in mortality in the conclusion. 

Author Response

The authors have to highlight the limitations of their study and must include possible causes for the decrease in mortality in the conclusion.

Thank you for your comments. We extended the limitations section. “We acknowledge several limitations. Although our results are consistent with current literature, this was a single-center retrospective study, and these results may not be fully transposable elsewhere, as our cohort reflects the case mix of cancer patients followed up in our institution. In addition, due to its retrospective design, some characteristics were missing and others could not be reliably collected, including the dose and duration of RABs treatment especially after hospital discharge, the relevant oncologic outcomes as well as the primary causes of death. Furthermore, this study analysed patients hospitalized over a 14-year period, during which time patients’ characteristics, ICU management and prognosis have probably evolved. Finally, despite consistent results in current literature, association is not causation so we cannot fully conclude on the benefit of these treatment in patients with malignancies.”

We discussed the possible causes for the improved survival in the conclusion section: “We make the assumption that it might be linked to antiproliferative effect or decreased cancer-treatment adverse events, although it has to be explored in prospective studies.”

We also added this sentence in the discussion section: “It is noteworthy that RABs users, compared to non-RABs users, displayed different baseline cancer characteristics with less advanced malignancies, which could affect long-term prognosis. Nevertheless, we used a propensity-score matching method to compare those patients.”

Reviewer 4 Report

The study aimed to evaluate the survival outcomes of cancer patients admitted to the intensive care unit (ICU) who were treated with renin-angiotensin system blockers (RABs). A retrospective observational study was conducted in a 24-bed medical ICU, including adult patients with solid tumors. Out of 1845 patients admitted with solid malignancies, RABs were used in 414 patients, either as angiotensin-receptor blockers (ARBs) or angiotensin-converting enzyme inhibitors (ACEi). After adjusting for other factors, the use of ARBs and ACEi was associated with improved in-ICU survival. Additionally, treatment with ARBs was independently associated with decreased one-year mortality, while treatment with ACEi did not show a similar association. The study focused on cancer patients admitted to the ICU, a population that often faces significant challenges and limited survival rates. The study considered both in-ICU survival and one-year survival, providing a broader understanding of the impact of RABs. Although study design is retrospective, which may introduce bias and limit the ability to establish a causal relationship between RABs and improved survival but sometimes can provide valuable insights based on real-world data and a large sample size.

The introduction provides a clear and concise overview of renin-angiotensin blockers (RABs) and their cardiovascular protective properties.It highlights the controversial nature of the impact of preadmission RABs on critically ill patients, setting the stage for the need for further investigation. The introduction references previous studies that have shown no increased mortality and potential survival benefits of RABs in ICU populations and ICU survivors with acute kidney injury. It acknowledges the growing proportion of cancer patients in ICU admissions and the potential benefits of RABs in reducing cancer treatment-related adverse events and inhibiting tumorigenesis. The objective of the present study is clearly stated, providing a clear direction for the research.

The materials and methods section provides a clear description of the study design, including its retrospective observational nature and single-center setting. The inclusion and exclusion criteria are well-defined, ensuring that the study focuses on adult patients with solid tumors requiring unplanned ICU admissions. The section outlines the data collection process, including demographic characteristics, comorbidities, prior chronic medications, and details related to the underlying malignancy and recent oncological treatment. The recording of relevant ICU-related data, such as admission scores, organ failure supports, and decisions to forgo life-sustaining therapies, is comprehensive and important for understanding patient outcomes. The statistical analysis plan is detailed, describing the univariate and multivariate analyses, logistic regression, propensity score weighting, and survival analysis techniques used. The approval of the study by the ethics committee is mentioned, demonstrating adherence to ethical guidelines. Although the study is described as retrospective, there is no mention of how data were collected or the sources of the data, which could impact the reliability and generalizability of the findings. The section does not specify how missing data or data quality issues were handled, which is important for ensuring the validity of the results.

The results starts by clearly stating the total number of patients included in the study, providing a basis for understanding the following results. Baseline characteristics of the cohort, including age, gender, primary tumor sites, metastatic stage, and ICU admission features, are presented in a concise tabular format (Table 1 and Table 2), allowing for easy reference and comparison. The use of angiotensin-receptor blockers (ARBs) and angiotensin-converting enzyme inhibitors (ACEi) is examined separately, providing insights into the effects of these medications on various outcomes. Multivariate analyses are performed to adjust for confounding factors and identify independent determinants of ICU mortality, in-hospital mortality, and one-year mortality. Survival curves (Figure 1, Figure 2, Figure 3) are included to visually illustrate the differences in outcomes based on the use of ARBs or ACEi. The section concludes with a discussion of the results and provides additional information on the effects of ARBs on one-year mortality through propensity score matching.

The discussion section begins by summarizing the main findings and highlighting the improved survival in the ICU and one-year survival among patients treated with RABs (ARBs and ACEi). The authors discuss the potential mechanisms underlying the observed effects of RABs on tumor progression, such as the renin-angiotensin system pathway and its impact on tumor proliferation, angiogenesis, and apoptosis. They provide examples from preclinical studies and clinical investigations that support the hypothesis of RABs as adjunctive therapy in cancer treatment, including improvements in chemotherapy efficacy and mitigation of treatment-related adverse events. The authors acknowledge the limitations of the study, such as its retrospective design and the lack of certain important data, and highlight the need for further large, randomized studies to explore the effects of RABs in cancer patients. The conclusion reiterates the key finding of improved one-year survival among ARBs users and emphasizes the lack of association between ACEi use and one-year mortality in this specific population. The section would benefit from more thorough exploration and discussion of the limitations mentioned, such as the retrospective design, missing data, and potential biases, in order to provide a clearer understanding of the study's limitations and their impact on the results.

Overall, the Discussion section provides a reasonable interpretation of the study's findings and discusses the potential implications of the results. However, further expansion of limitations and alternative explanations, would strengthen the discussion.

Top of Form

Bottom of Form

Author Response

We thank you for you careful review of our study. We modified the manuscript as required.

In the materials and methods section, we added the following sentence: “Data were extracted from patient's data management system (Clinisoft®, GE Healthcare) and computed from the individual medical file”. “Missing data were handled using the Random Forest method”.

In the discussion section, we added this sentence: “It is noteworthy that RABs users, compared to non-RABs users, displayed different baseline cancer characteristics with less advanced malignancies, which could affect long-term prognosis. Nevertheless, we used a propensity-score matching method to compare those patients.”

We also extended the limitations paragraph: “We acknowledge several limitations. Although our results are consistent with current literature, this was a single-center retrospective study, and these results may not be fully transposable elsewhere, as our cohort reflects the case mix of cancer patients followed up in our institution. In addition, due to its retrospective design, some characteristics were missing and others could not be reliably collected, including the dose and duration of RABs treatment especially after hospital discharge, the relevant oncologic outcomes as well as the primary causes of death. Furthermore, this study analysed patients hospitalized over a 14-year period, during which time patients’ characteristics, ICU management and prognosis have probably evolved. Finally, despite consistent results in current literature, association is not causation so we cannot fully conclude on the benefit of these treatment in patients with malignancies.”

We underlined the limitations adding this sentence in the conclusion section, as required by reviewer 3: “We make the assumption that it might be linked to antiproliferative effect or decreased cancer-treatment adverse events, although it has to be explored in prospective studies.”